# Pioneering the Equation of State of Dense Nuclear Matter with Strange Particles Emitted in Heavy-Ion Collisions: The KaoS Experiment at GSI

Peter Senger [1,2]

1    Facility for Antiproton and Ion Research, 64291 Darmstadt, Germany; p.senger@gsi.de
2    National Research Nuclear University MEPhI, 115409 Moscow, Russia

**Abstract:** High-energy heavy-ion collisions offer the unique possibility to study fundamental properties of nuclear matter in the laboratory, which are relevant for our understanding of the structure of compact stellar objects and the dynamics of neutron star mergers. Of particular interest are the nuclear matter equation of state (EOS), the in-medium modifications of hadrons and the degrees of freedom of matter at high densities and temperatures. Pioneering experiments exploring the EOS for symmetric matter were performed at the SIS18 accelerator of GSI, measuring, as function of beam energy, the collective flow of protons and of light fragments and subthreshold strangeness production. These data were reproduced by various microscopic transport model calculations, providing, up to date, the best constraint for the EOS of symmetric matter with an incompressibility of about 200 MeV for densities up to twice the saturation density. This article reviews the experimental results on subthreshold kaon production together with the theoretical interpretation and gives a brief outlook towards future experiments at higher densities.

**Keywords:** heavy-ion collisions; strange particle production; nuclear matter equation of state



## 1. Introduction

About 4 decades ago, experiments with beams of heavy nuclei such as Au and Pb started with the motivation to investigate the bulk properties of dense nuclear matter. In collisions between these heavy nuclei at bombarding energies, in the order of several hundred MeV per nucleon up to 1A GeV, it was expected that the nucleons piled up in the reaction volume to densities of up to 2–3 times the saturation density $\rho_0$. The big question was whether the nucleons would be stopped and form a new kind of nuclear matter, or whether the nucleons would just undergo binary inelastic collisions. First, Au beams at this energy became available at the LBL in Berkeley in the early 1980s. The Plastic Ball collaboration found experimental evidence for proton pile-up at midrapidity and discovered an anisotropic emission pattern of protons in respect to the reaction plane, which was interpreted as a collective flow phenomenon [1]. The Streamer Chamber collaboration at LBL measured the excitation function of pion production in La+La collisions and estimated, from the pion yields, the fireball temperature and the potential part of the compressional energy, indicating a stiff nuclear matter equation of state (EOS) [2]. However, when taking into account temperature- and density-dependent pion in-medium effects, the pion yields could be also reproduced assuming a soft EOS [3]. The observation of a collective flow in heavy-ion collisions was confirmed by experiments at the SIS18 accelerator at GSI, which delivered the first Au beams in 1990. The collective flow of both protons and neutrons was measured in Au+Au collisions at 400, 600 and 800A MeV using a combination of the Four Pi (FOPI) detector and the Large Neutron Detector (LAND) [4]. In experiments with the kaon spectrometer (KaoS), the production of strange particles was observed for the first time in Au+Au collisions [5,6], together with the discovery of the elliptic flow kaons [7]. In the new millennium, the High Acceptance Dielectron Spectrometer (HADES) was installed

and commissioned at GSI/SIS18, with the goal to perform systematic investigations of the production of electron–positron pairs and hadrons in pion-, proton- and heavy-ion-induced reactions [8]. One of the outstanding experimental results is the direct measurement of the average fireball temperature in Au+Au collisions at a beam energy of 1.23A GeV [9].

Higher beam energies were reached at the AGS facility in Brookhaven, where the first measurements with a Au beam at 11.6A GeV/c were performed in 1992 by the E866 and E877 collaborations. In addition, at these energies, a substantial amount of stopping and a high degree of thermalization were observed in central Au+Au collisions [10]. The E895 collaboration measured the excitation function of proton-directed and elliptic flow in Au+Au collisions from 2–8A GeV [11]. The interpretation of these data by microscopic transport calculations provided a constraint of the high-density EOS, which still serves as a benchmark for model calculations [12]. In the mid-1990s, Pb beams also became available at the CERN SPS and several experiments started to search indications for the creation of the quark–gluon plasma (QGP) in heavy-ion collisions using different observables; the CERES/NA45 collaboration observed a significant excess of the $e^+e^-$ pair yield over the expectation from hadron decays in Pb+Au collisions at 158A GeV/c, an effect which was attributed to the in-medium broadening of the $\rho$-meson spectral function [13]. The NA50 collaboration measured muon pairs as a function of centrality in Pb+Pb collisions at 158A GeV/c and found a strong decrease in the charmonium yield for central collisions, which was interpreted as anomalous $J/\Psi$ suppression in the QGP [14]. The excitation function of charged hadron production was studied in the NA49 experiment in Pb+Pb collisions for beam energies starting from 20A GeV up to 158A GeV. The observed $K^+/\pi^+$ ratio exhibited a peak at a beam energy around 30A GeV, an effect which was regarded as a signature for the "onset of deconfinement" [15]. The measured particle yields can be reproduced by statistical models assuming a freeze-out temperature T and a baryon chemical potential $\mu_B$ for each energy, resulting in the "freeze-out curve" in the QCD phase diagram T versus $\mu_B$ [16–18].

Since 2000, the Relativistic Heavy-Ion Collider (RHIC) at BNL has provided Au+Au collisions at a top energy of $\sqrt{s_{NN}}$ = 200 GeV, which have been investigated by several experiments. The major discoveries, which shed light on the creation and the properties of a new state of deconfined matter, the QGP, include the suppression of high-energetic particles ("jet quenching") [19] and the constituent quark number scaling of elliptic flow of particles [20]. Ten years later, in 2010, Pb beams finally also became available at the LHC at CERN, starting with collision energies of $\sqrt{s_{NN}}$= 2.76 and, later, 5.02 TeV. In collisions at this ultra-relativistic beam energy, the number of created antiparticles equals the number of particles. Under this condition, i.e., at vanishing baryon chemical potential, lattice QCD calculations find a smooth chiral crossover from the quark–gluon plasma to hadronic matter at a pseudocritical temperature of $T_{pc}$ = 155–160 MeV [21,22]. A consistent value was extracted for the freeze-out temperature from the particle yields measured by the ALICE collaboration at the LHC using a statistical hadronization model [23]. This finding suggests that, in heavy-ion collisions at ultra-relativistic energies, the particle freeze-out coincides with the chiral crossover phase change. However, for larger values of baryon chemical potential, model calculations predict a first-order phase transition with a critical endpoint [24].

In order to search systematically for this critical endpoint, the STAR collaboration has performed a beam energy scan at RHIC. Part of this program was the search for the disappearance of QGP signals when lowering the beam energy such as the constituent quark number scaling of the elliptic particle flow [25]. Indications of the breakdown of this scaling behavior were found at $\sqrt{s_{NN}}$ = 3 GeV, an energy which could only be reached in the fixed target operation mode of the STAR experiment [26]. Another important goal of the beam energy scan was to search for signatures of a critical endpoint, such as a non-monotonic variation in the moments of the proton multiplicity distribution as function of beam energy, which is related to the correlation length and the susceptibilities of the system. An indication for this effect was found at $\sqrt{s_{NN}}$ = 7.7 GeV, the lowest collision

energy measured in the RHIC collider mode [27]. Recent lattice QCD calculations found that the temperature of a hypothetical critical point of a chiral phase transition should not exceed a value of $T_c = 132 + 3 - 6$ MeV [28,29]. QCD-inspired model calculations locate the critical endpoint at a temperature of $T_{cep} = 93$ MeV and at a baryon chemical potential of $\mu_{cep} = 672$ MeV [24]. Such conditions are reached in collisions with fixed targets at STAR [26] and will be reached at the future fixed-target CBM experiment at FAIR and at the MPD experiment at the NICA collider [30]. These experiments, together with the fixed-target BM@N experiment at JINR, will also contribute to the exploration of the high-density EOS, hence closing the gap to the FOPI, KaoS and HADES experiments at GSI/SIS18.

This article is devoted to the experimental results obtained with the kaon spectrometer at GSI. This experiment pioneered subthreshold strangeness production in heavy-ion collisions and provided a data set which allowed the extraction of information on the high-density EOS of symmetric nuclear matter and on the in-medium modifications of particles. The KaoS data, together with the flow data of FOPI, still represent the benchmarks for transport models used for the extraction of the EOS up to densities of $2\,\rho_0$ [31].

## 2. The Kaon Spectrometer (KaoS) Experiment at GSI/SIS18

In the late 1980s, a double-focusing QD magnetic spectrometer was developed and installed at the SIS18 heavy-ion facility at GSI in Darmstadt [32]. The spectrometer's primary purpose was to study meson production in energetic nucleus–nucleus collisions. Its compact design was matched to the requirements of kaon detection with a short flight path (5–6.5 m), a large solid angle (up to 35 msr), a wide momentum acceptance ($p_{max}/p_{min} = 2$), a maximum momentum of 1.6 GeV/c (1.9 GeV/c at reduced solid angle) and a reasonable momentum resolution (=1% without and about $10^{-3}$ with track reconstruction). Figure 1 depicts a sketch of the setup in the left panel, together with a photo in the right panel. A focal plane length of about 1.5 m allows the efficient use of the detectors necessary for track reconstruction and particle identification, involving wire chambers, time-of-flight scintillators and Cherenkov detectors, to be carried out. Track reconstruction is based on three large-area multi-wire proportional counters (MWPC 1–3), one between the quadrupole and the dipole and two behind the dipole magnet, each of them measuring two spatial coordinates. The efficiencies for kaon detection are larger than 95% for each of these detectors.

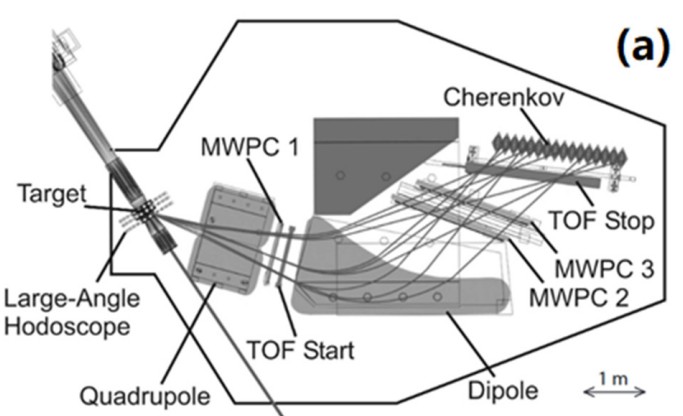
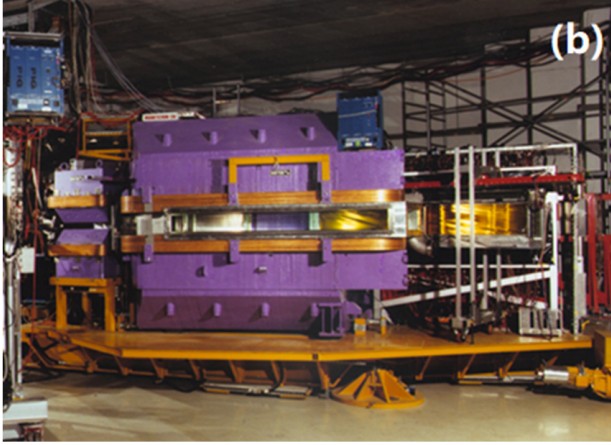

**Figure 1.** (**a**) Layout of the double-focusing magnetic spectrometer KaoS with its detector system: time-of-flight (TOF) start and stop detectors, 3 multi-wire proportional chambers (MWPC), Cherenkov detectors with water, lucite and silica aerogel radiators, two hodoscopes for event characterization at target (for large angles) and 7 m downstream the target (for small angles, not shown) [32]. (**b**) Photo of the setup.

The time-of-flight (TOF) is measured with segmented plastic scintillator arrays. The TOF start detector consists of 16 modules and is located between the quadrupole and the dipole, while the TOF stop detector comprises 30 modules arranged along the focal plane of the spectrometer. Particle identification is based on momentum determination using the MWPCs and on time-of-flight (TOF) measurements. For the separation of high momentum protons from kaons, a threshold Cherenkov detector is used in addition. The kaon trigger is based on time-of-flight information, which is able to suppress pions and protons by factors of 100 and 1000, respectively. Collisions can be characterized by two multiple-module plastic-scintillator hodoscopes detecting reaction fragments in the forward hemisphere; the large-angle hodoscope (LAH) around the target point consists of 96 modules and covers polar laboratory angles of $12° \leq \theta\text{lab} \leq 48°$, while the small-angle hodoscope (SAH) is located 7 m downstream the target, covers an active area of $2.24 \times 192$ m$^2$ and comprises 380 modules of increasing size from the center to the outer region ($4 \times 4$ cm$^2$, $8 \times 8$ cm$^2$ and $16 \times 16$ cm$^2$). Positively and negatively charged particles are measured separately using different magnetic field polarities. The spectrometer is mounted on a platform which can be rotated around the target point on air cushions in a polar angel range from $\theta_{\text{lab}} = 0°$ to $130°$. The angular range covered at each position is $\Delta\theta_{\text{lab}} = \pm 4°$. The beam intensity is monitored using two scintillator telescopes positioned at backward angles ($\theta_{\text{lab}} = \pm 110°$), measuring the flux of charged particles produced in the target which is proportional to the beam intensity. The absolute normalization is obtained in separate measurements at low beam intensities using a plastic scintillation detector directly in the beam line. The beam intensities were chosen such that the dead time of the data acquisition system (DAQ) is always below 50%. The momentum coverage is maximized by measuring different magnetic field settings ($|B_{\text{dipole}}| = 0.6, 0.9$ and $1.4$ T). The resulting acceptance for kaons as function of normalized rapidity $y/y_{\text{beam}}$ and of transverse momentum is shown in Figure 2 for three different beam energies (1.0, 1.5 and 1.93A GeV). The shaded areas correspond to different angular settings $\theta_{\text{lab}}$ of the spectrometer in the laboratory, as denoted in the figure, and to various magnetic field settings. More details on the experiment can be found in [32].

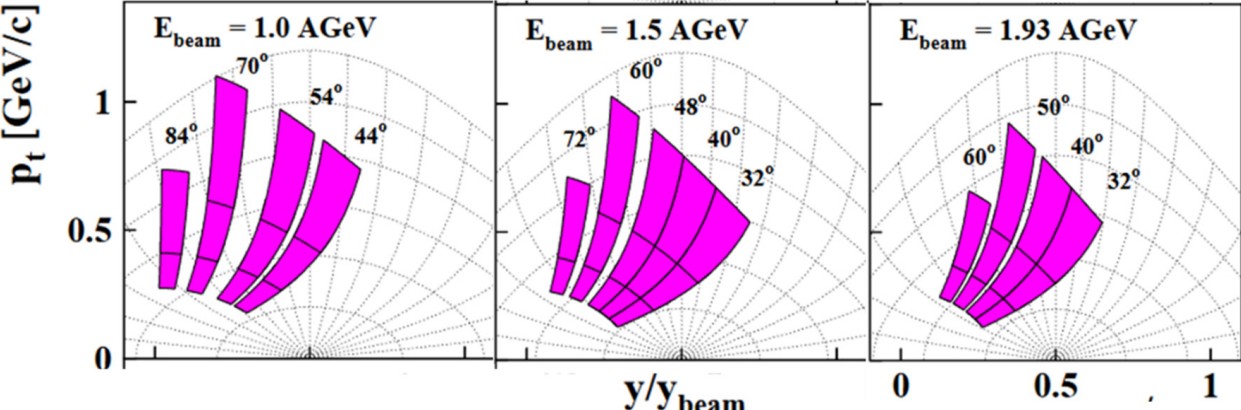

**Figure 2.** Examples of the acceptance of the kaon spectrometer in the plane transverse momentum versus normalized rapidity $y/y_{\text{beam}}$ for several laboratory angles $\theta_{\text{lab}}$ (as indicated) and for various magnetic field settings at 1.0A GeV, at 1.5A GeV and at 1.93A GeV beam energy. The angular acceptance is $\Delta\theta_{\text{lab}} = \pm 4°$, corresponding to the width of the pink bands.

## 3. Experimental Results

In the following, the data on pion and kaon production in nuclear collisions as measured with the kaon spectrometer at GSI-SIS18 are reviewed.

### 3.1. Pion Production

In high-energy collisions between nuclei, the nucleons of projectile and target pile up in the reaction volume. A substantial part of the kinetic energy of the nucleons is dissipated

into compressional energy, chaotic motion, i.e., thermal energy and intrinsic excitation of the nucleons. The pressure built up in the dense and hot "fireball" drives the system apart leading to a collective flow of matter. The fireball temperature is converted into kinetic energy of the emitted particles. The excited nucleons, i.e., short-lived baryonic resonances—at SIS18 beam energies, mostly Δ resonances—subsequently decay again by emission of pions, which might be reabsorbed again. Figure 3 depicts the number of pions per participating nucleon as a function of the available energy in a nucleon–nucleon collision, measured in various nucleus–nucleus (symbols). The red line corresponds to pion data from nucleon–nucleon (N+N) collisions, which are extracted from pion production in proton–proton collisions by correction for isospin effects. In the GSI/LBL beam energy range, the pion yield per participating nucleon measured in N+N collisions clearly exceeds the corresponding heavy-ion data. This is due to the fact that, in inelastic nucleon–nucleon collisions, the dissipated energy is fully converted into Δ excitations and, finally, into pion production, as no compressional and thermal energy is lost for the creation of a fireball. Moreover, no pion is reabsorbed in nucleon–nucleon collisions. Similarly, the pion yield per participating nucleons is higher for light collisions systems such as C+C than Au+Au, as pion reabsorption is reduced and less energy goes into compression and heat. With increasing beam energies, multiple collisions in nucleus–nucleus collisions also happen more abundantly and pion production increasingly exceeds the values for nucleon–nucleon collisions.

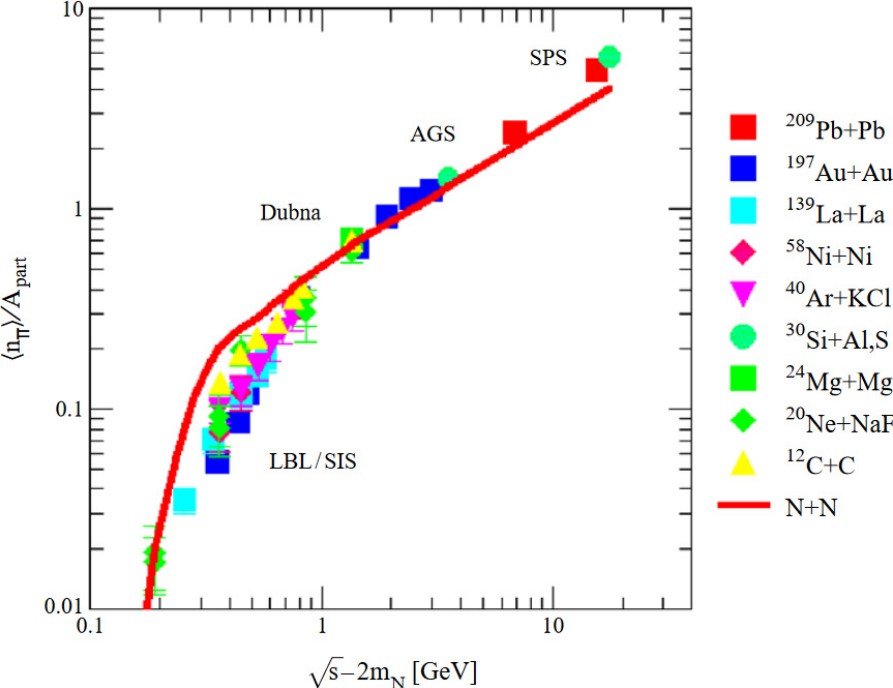

**Figure 3.** Pion multiplicity per participating nucleon measured in nucleus–nucleus collisions (symbols) and in nucleon–nucleon collisions as a function of available energy in the NN system (taken from [33]).

More information on pion production in heavy and light collision system can be extracted from Figure 4, which depicts differential pion production cross sections as function of kinetic energy in the c.m. system for different polar emission angles. The left and center panels depict $\pi^+$ and $\pi^-$ spectra from Au+Au collisions at a beam energy of 1.5A GeV, respectively, whereas, in the right panel, $\pi^+$ spectra from C+C collisions at 1A GeV are shown [34]. The data were measured by the KaoS collaboration and are compared to results of calculations with the UrQMD event generator. For the heavy system, data and model calculations differ in both yield and spectral slope. While the measured soft pion yield exceeds the UrQMD results, the yield of hard pions is overestimated by the model

calculations. These discrepancies reflect the difficulty in describing properly the lifetime and spectral function of the $\Delta$-resonance and the $\Delta N$ cross-section in the dense and hot media, as created in Au+Au collisions.

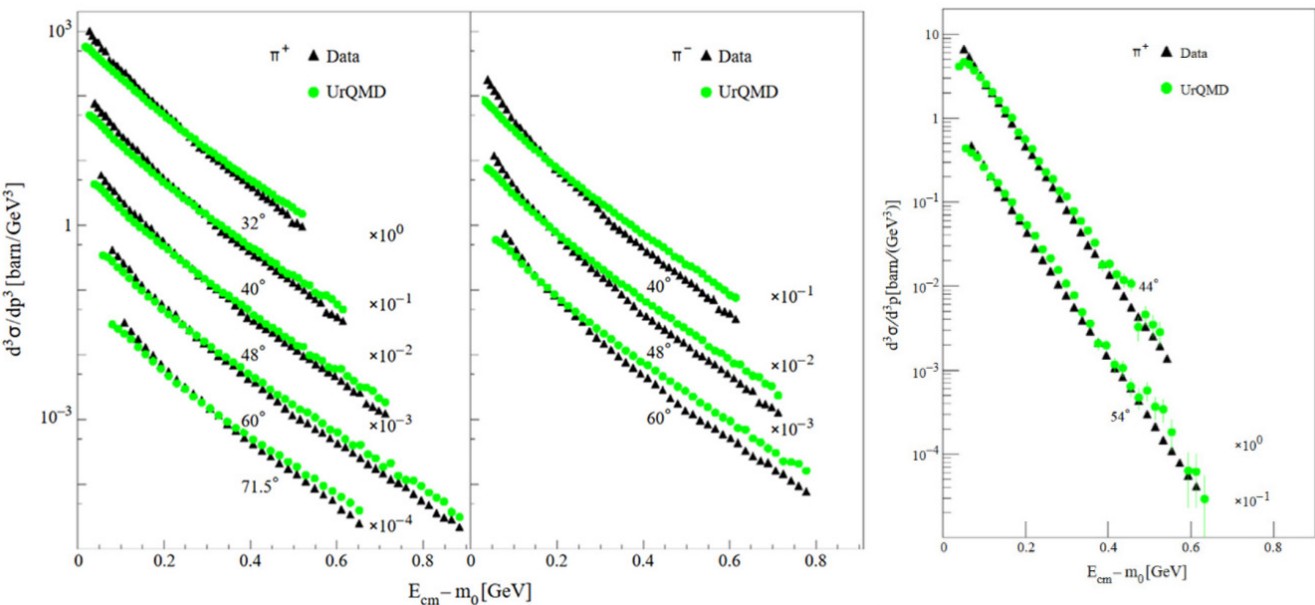

**Figure 4.** Differential pion production cross-sections measured at various polar emission angles in Au+Au collisions at 1.5A GeV ($\pi^+$ left panel and $\pi^-$ center panel) and in C+C collisions at 1A GeV ($\pi^+$ right panel) as a function of the kinetic energy in the c.m. system (black triangles) in comparison to results of URQMD transport calculations (green triangles) [34].

Figure 4 also demonstrates that the pion spectra measured in Au+Au collisions deviate from a single Boltzmann distribution. As the spectral slope of the pions reflects both the temperature of the fireball and its expansion, which vary over the collision time, the slope contains information from different stages of the collision, as the pions are emitted continuously during the fireball lifetime. The pion emission time and its relation to the pion energy have been investigated by the KaoS collaboration by measuring angular emission pattern of pions in semi-central Au+Au collisions [35]. If both pions and fragments are emitted in the same direction, the pions are shadowed by the spectator fragments. This situation is illustrated in the left panel of Figure 5, which depicts three snapshots of semi-central Au+Au collisions at a beam energy of 1A GeV as calculated with a QMD model. In the experiment, pions were selected which were emitted into the reaction plane ($\pm 45°$) near target rapidity ($\theta_{lab} = 84°$), as indicated by the arrows in the left panel of Figure 5. The reaction plane is defined for each event by the vector sum of the transverse momenta of all spectator particles measured in the small-angle hodoscope within $0.5° \leq \theta_{lab} \leq 5°$ (see above).

The result of the measurement is shown in the right panel of Figure 5, which depicts the ratio $N^\pi_{proj}/N^\pi_{targ}$ as a function of the pion transverse momentum, where $N^\pi_{proj}$ and $N^\pi_{targ}$ are the numbers of pions emitted to the projectile and to the target side within a cone of polar angles $\Delta\varphi = \pm 45°$, respectively [35]. The data indicate that the ratio $N^\pi_{proj}/N^\pi_{targ}$ decreases with the increase in the pion transverse momentum. This observation suggests that pions are re-scattered or absorbed by spectator fragments. Pions emitted in the early stage of the collision towards target rapidity can interact only with the projectile fragment, but not with the target fragment, as illustrated by the left snapshot in Figure 5, which was taken 4 fm/c after the first touch of the nuclei. The resulting ratio $N^\pi_{proj}/N^\pi_{targ}$ is smaller than unity, which is the case for pions with large transverse momenta (see right panel of Figure 5). However, for pions with small transverse momenta, the ratio $N^\pi_{proj}/N^\pi_{targ}$ is above unity, indicating that these soft pions are shadowed by the target spectator in

the late stage of the collision, as illustrated by the right snapshot taken at 16 fm/c. In conclusion, the experimental results confirm the picture whereby hard pions leave the fireball early, whereas soft pions freeze out late. This observation is supported by statistical hadronization models, which reproduce the total particle yields in heavy-ion collisions assuming freeze-out conditions. This finding also applies to the pion yields, which are dominated by soft pions. On the other hand, it should be mentioned that theoretical models predict, for soft quasi-pions, a larger path length in nuclear matter than for hard pions. In this case, soft pions would leave the fireball earlier than hard pions [36].

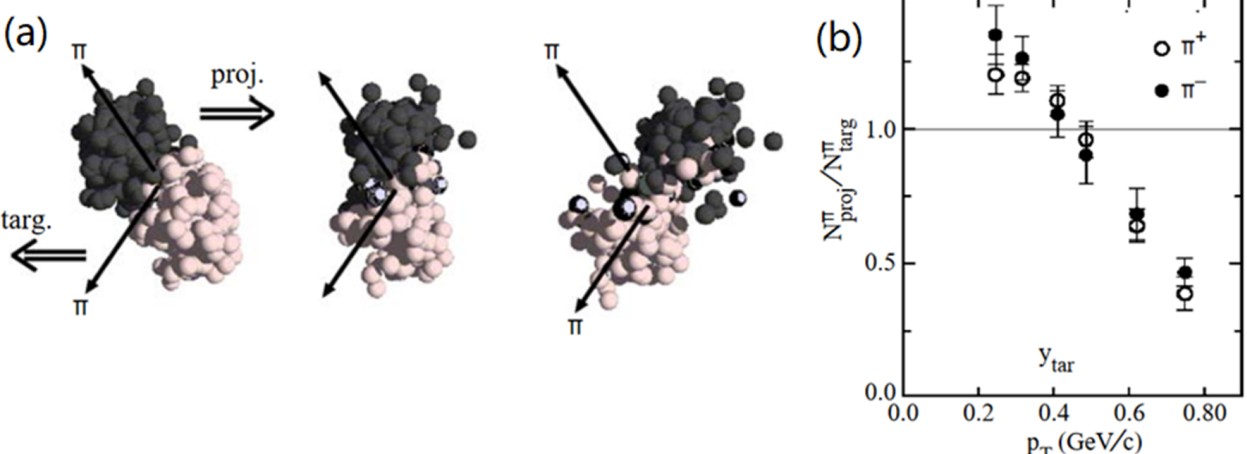

**Figure 5.** (**a**) Three snapshots of a Au+Au collision with a beam kinetic energy of 1A GeV (impact parameter b = 7 fm) calculated with the QMD transport code for 4 fm/c (left), 10 fm/c (center) and 16 fm/c (right). The pions are emitted in the reaction plane at backward angles corresponding to a particular detector position. (**b**) Pion number ratio $N^\pi_{proj}/N^\pi_{targ}$ measured as function of transverse momentum in peripheral Au+Au collisions (b ≥ 5.7 fm) at 1A GeV at target rapidity. $N^\pi_{proj}$ and $N^\pi_{targ}$ denote the numbers of pions emitted to the projectile and to the target side, respectively, within a cone of ±45° [35].

The pion emission pattern perpendicular to the reaction plane also was analyzed for the first time by the KaoS collaboration. The left panel of Figure 6 depicts the azimuthal distributions of $\pi^+$ mesons emitted in peripheral, semi-central and central collisions (from top to bottom) measured in Au+Au collisions at 1A GeV [37]. The values $\varphi = \pm90°$ and $\varphi = \pm180°$ correspond to the emission angles perpendicular and parallel to the reaction plane, respectively. The pions were analyzed for two momentum ranges, i.e., $160 < p_T < 260$ MeV/c (left row) and $260 < p_T < 600$ MeV/c (right row). The solid lines represent fits to the data. In particular, for semi-central collisions (middle row) the pion polar angle distributions exhibit clear peaks at $\varphi = \pm90°$. This pattern is more pronounced for hard pions, which are emitted early, when the projectile and target spectator still act efficiently as shadow. The effect is even more strongly visible for peripheral collisions (top row), when the hard pions are shadowed by larger spectator fragments. For central collisions (lower row), the azimuthal emission pattern is washed out because the spectator fragments vanish. The right panel of Figure 6 sketches the geometry of a semi-central heavy-ion collision, illustrating the shadowing effect by the projectile and target spectator fragments, which has also been observed for protons [1]. In addition, for $\pi^0$ mesons, an enhanced emission perpendicular to the reaction plane was found in Au+Au collisions at 1A GeV by the TAPS collaboration at GSI [38].

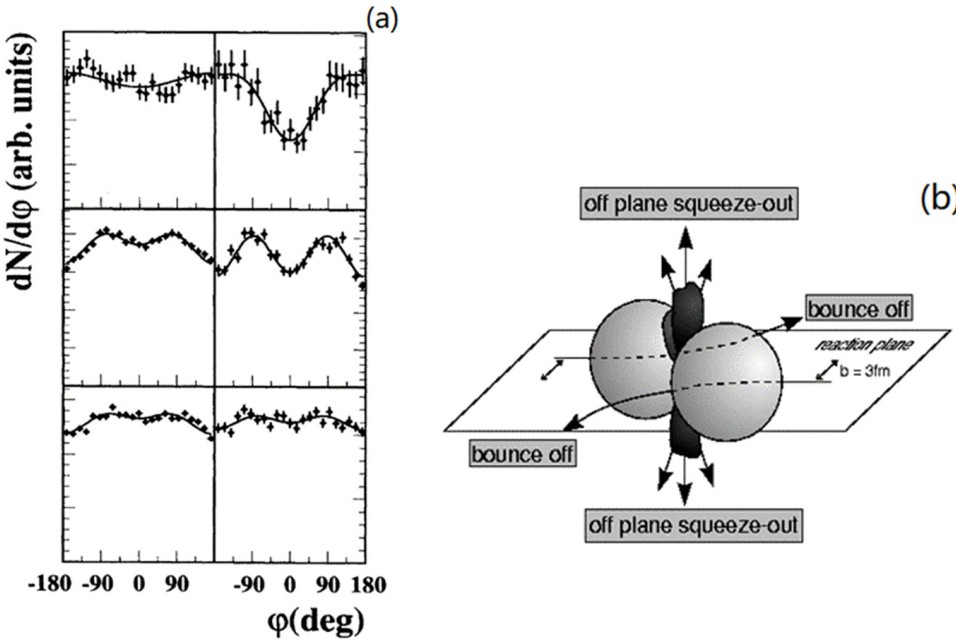

**Figure 6.** (**a**) Azimuthal distributions of positively charged pions for peripheral, semi-central and central collisions (from top to bottom) measured in Au+Au collisions at 1A GeV [37]. The ordinate is linear starting at zero. Left column: $\pi^+$ in the range $160 < p_T < 260$ MeV/c. Right column: $\pi^+$ in the range $260 < p_T < 600$ MeV/c. Solid lines: fits to the data with cos ($\varphi$) and cos ($2\varphi$) terms. $\varphi = 0°$ and $\varphi = \pm180°$ represent emission of pions parallel to the reaction plane and $\varphi = \pm90°$ corresponds to emission of pions perpendicular to the reaction plane. (**b**) Illustration of the particle emission pattern for semi-central collisions at intermediate beam energies perpendicular to the reaction plane ("off-plane squeeze-out") and parallel to the reaction plane ("bounce off").

*3.2. Kaon Production*

The production of strange particles in heavy ion collisions at SIS18 energies is a very promising probe of the collision dynamics and the matter properties in the reaction volume, as the available beam energies per nucleon are mostly below the threshold energy for strangeness production in nucleon–nucleon collisions. The creation of a $K^+$ meson via the process p + p → $K^+$ $\Lambda$ p requires a proton kinetic energy of 1.58 GeV and $K^-$ mesons can be produced via p + p → $K^+K^-$ pp above a proton energy of 2.5 GeV. A parameterization of the excitation function of $K^+$ and $K^-$ mesons per participating nucleon in nucleon–nucleon collisions is presented in Figure 7 as a blue line and a red dashed line, respectively, as functions of the Q-value, i.e., the available energy above the threshold. The open blue squares and the open cyan circles indicate the $K^+$ yields per participating nucleons measured in C+C and Ni+Ni collisions, respectively, while the full read squares and the full green dots show the corresponding yields for $K^-$ mesons [39,40]. Figure 7 demonstrates that kaon production in heavy-ion collisions at SIS18 energies requires processes in addition to binary nucleon–nucleon collisions. According to microscopic transport models, the production of strange particles in nucleus–nucleus collisions at subthreshold beam energies proceeds via multi-step processes in the reaction volume, involving $\Delta$ resonances, N* and pions, such as $\pi$ p → $K^+\Lambda$ $\pi$.

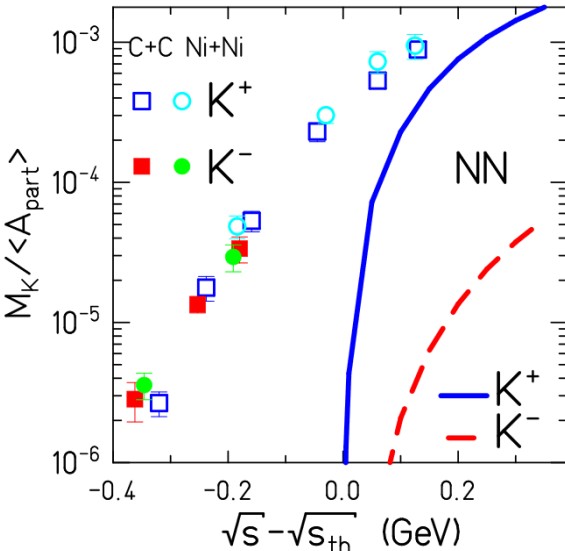

**Figure 7.** K$^+$ and K$^-$ multiplicity per number of participating nucleons as a function of the available energy above threshold in first-chance collisions for C+C and Ni+Ni collisions (for symbols, see legend) and parameterizations of the kaon production cross sections in nucleon–nucleon collisions (for lines, see insert). Taken from [40].

A selection of KaoS results on kaon production measured for different collision systems and beam energies are displayed in Figure 8. The invariant production cross-sections $\sigma_{inv} = E\, d^3\sigma/dp^3$ for K$^+$ mesons (left panel) and K$^-$ mesons (right panel) at mid-rapidity are plotted as functions of the kinetic energy in the c.m. system $E_{c.m.} - m_0c^2$. These "mid-rapidity distributions" were produced by selecting data points within $\theta_{c.m.} = 90° \pm 10°$ from measurements at various laboratory angles. The lines are Maxwell–Boltzmann distributions $E\, d^3\sigma/dp^3 \sim E_{c.m.}\, \exp(-E_{c.m.}/T)$ fitted to the data with T the inverse slope parameter [41]. The total K$^+$ and K$^-$ multiplicities measured in Au+Au, Ni+Ni and C+C collisions are presented in Figure 9, together with fits to the data [41].

The KaoS collaboration also investigated the azimuthal angular distribution of K$^+$ mesons and discovered a strong anisotropy. The results are presented in Figure 10, which depicts the K$^+$ azimuthal angular distribution measured in Au+Au collisions at 1A GeV [7]. The kaons were analyzed within a range of transverse momenta of $0.2\ \text{GeV}/c \leq p_t \leq 0.8\ \text{GeV}/c$ and for two normalized rapidity bins, $0.4 \leq y/y_{proj} \leq 0.6$ (left panel) and $0.2 \leq y/y_{proj} \leq 0.8$ (right panel). The K$^+$ distribution exhibits a peak around $\varphi = \pm 90°$, i.e., perpendicular to the reaction plane. Such an emission pattern was also observed for pions (see Figure 6), where it was attributed to the interaction with the spectator fragments. However, in contrast to the pions, the K$^+$ mean free path in nuclear matter is about 5 fm and rescattering at the spectators should be less important. This is illustrated by the dotted lines in Figure 10, which represent the results of transport calculations, taking into account K$^+$ rescattering only ([42]; left) and additional Coulomb effects ([43]; right). However, the data are well reproduced when taking into account a repulsive in-medium K$^+$N potential, as demonstrated by the solid lines.

As illustrated in Figure 10, experiments on strangeness production in heavy-ion collisions are well suited to investigate the in-medium properties of kaons for different matter densities. The behavior of kaons and antikaons in dense nuclear matter has been studied by various model calculations [44,45]. The models predict an attractive kaon nucleon (scalar) potential and a kaon nucleon vector potential, which is repulsive for kaons but attractive for antikaons. Consequently, the total K$^+$N in-medium potential is weakly repulsive, while the K$^-$N potential is strongly attractive. These potentials affect the propagation of K$^+$ and K$^-$ in nuclear matter; while K$^+$ mesons are repelled from the high-density regions, K$^-$ mesons are attracted. This effect results in a characteristic azimuthal emission pattern for K$^+$ mesons, as shown in Figure 10.

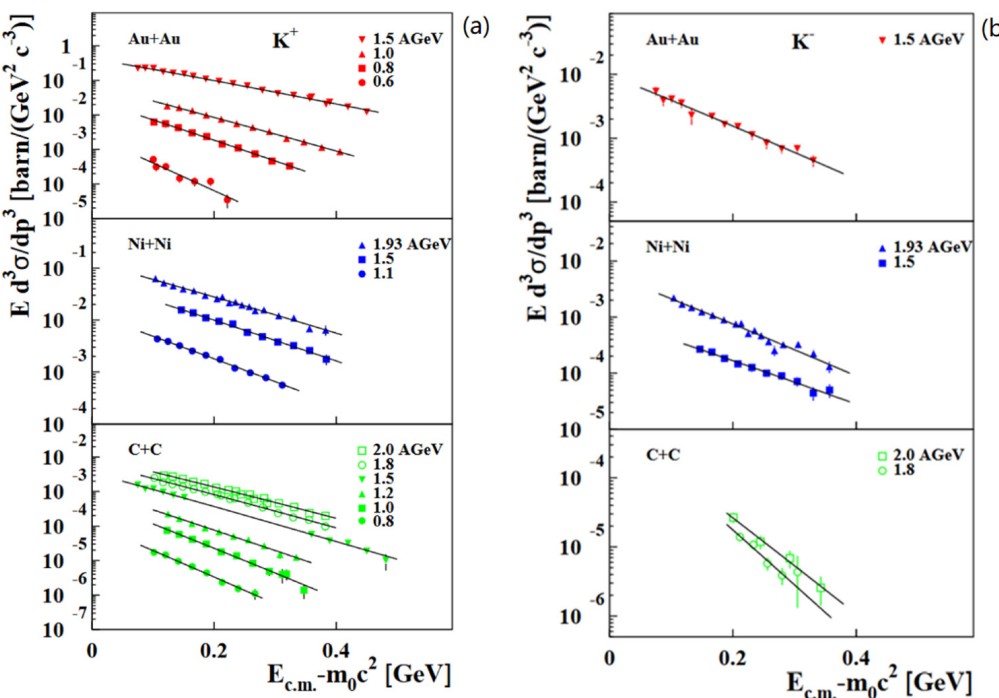

**Figure 8.** Inclusive invariant cross-sections at mid-rapidity as a function of the kinetic energy $E_{c.m.} - m_0c^2$ for $K^+$ mesons (**a**) and for $K^-$ mesons (**b**) for the various collision systems and beam energies measured. Mid-rapidity data were selected by the condition $\theta_{c.m.} = 90° \pm 10°$ from measurements at different polar angles [41]. The lines represent fits to the data (see text).

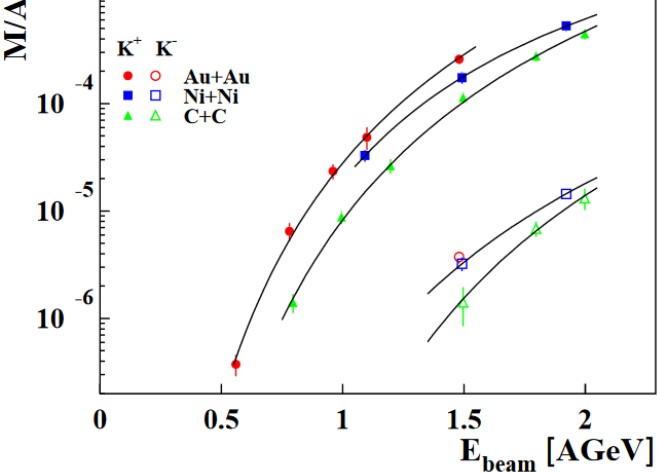

**Figure 9.** Multiplicities of $K^+$ (full symbols) and of $K^-$ mesons (open symbols) per mass number A of the collision system as a function of the beam energy. The lines represent fits to the data [41].

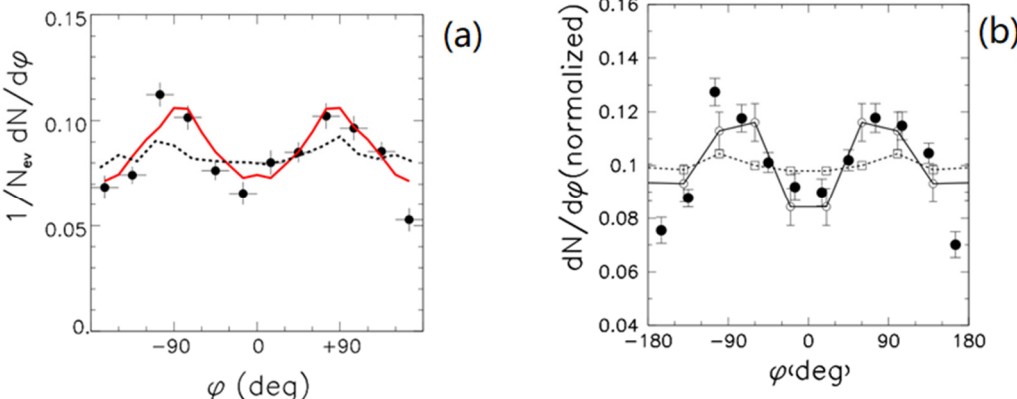

**Figure 10.** Azimuthal distribution of K$^+$ mesons measured in semi-central Au+Au collisions at 1A GeV (full dots). The kaons are analyzed for transverse momenta within a range of 0.2 GeV/c $\leq$ p$_t$ $\leq$ 0.8 GeV/c and for the normalized rapidity ranges of 0.4 $\leq$ y/y$_{proj}$ $\leq$ 0.6 (**a**) and 0.2 $\leq$ y/y$_{proj}$ $\leq$ 0.8 (**b**) [7]. The lines show the results of transport calculations using a RBUU model (left; [42]) and a QMD model (right; [43]), which both take into account rescattering; QMD also calculates Coulomb effects. Solid and dashed lines: calculations with and without in-medium K$^+$N potential, respectively. Taken from [40].

The beam energy of 1A GeV is too low to perform a high statistics measurement for K$^-$ mesons, the production of which requires 2.5 GeV in nucleon–nucleon collisions. Therefore, Ni+Ni collisions were performed at GSI with a beam energy of 1.93 A. The azimuthal angular distribution for $\pi^+$, K$^+$ and K$^-$ mesons measured by the KaoS collaboration in semi-central Ni+Ni collisions are depicted in Figure 11 [46]. The distributions of $\pi^+$ and K$^+$ mesons are similar but less pronounced than in Au+Au collisions, which is expected for smaller spectator fragments. However, the emission pattern of K$^-$ mesons exhibits peaks at $\pm$180°, corresponding to an in-plane elliptic flow. This effect was observed for the first time in collisions at SIS18 energies. Although K$^-$ mesons are expected to be strongly absorbed in spectator matter by strangeness exchange reactions such as K$^-$n $\rightarrow$ $\Lambda\pi^-$, their strongly attractive in-medium K$^-$N potential overcompensates absorption effects, resulting in an in-plane elliptic flow.

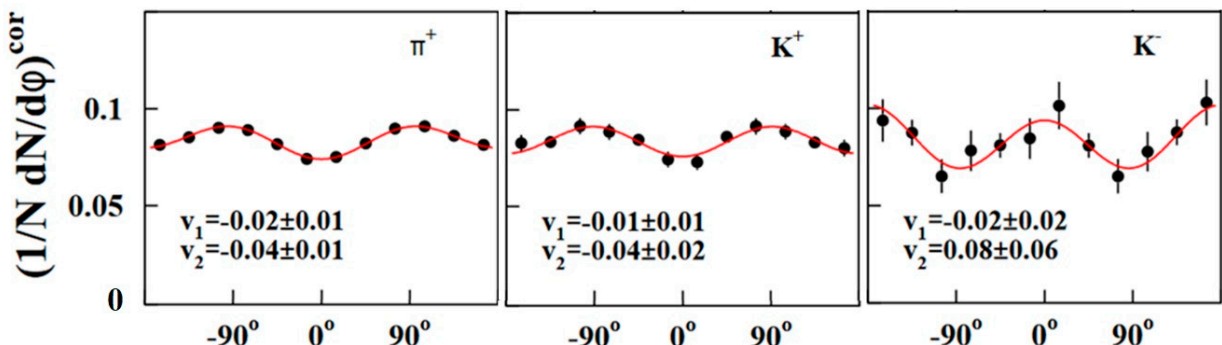

**Figure 11.** Azimuthal angular distributions of $\pi^+$, K$^+$ and K$^-$ mesons (from **left** to **right**) measured in semi-central Ni+Ni collisions at 1.93A·GeV [46]. The mesons are measured within a rapidity range of 0.3 < y/y$_{beam}$ < 0.7 and a momentum range of 0.2 GeV/c < p$_t$ < 0.8 GeV/c. The data are fitted using the first two components of a Fourier series dN/d$\Phi$~2 v$_1$ cos ($\varphi$) + 2 v$_2$ cos (2$\varphi$). The resulting values for v$_1$ and v$_2$ are indicated.

A further consequence of the in-medium KN potentials is a modification of the K$^+$ and K$^-$ effective mass in nuclear matter. According to model calculations [44], the slightly repulsive K$^+$N in-medium potential results in a K$^+$ effective mass moderately increasing with nuclear density, whereas the strongly attractive K$^-$N potential leads to a K$^-$ effective

mass, which considerably decreases with the increase in density. The latter effect may also result in a reduced K$^-$ absorption cross-section in nuclear matter. The in-medium mass modification of kaons manifests itself in the kaon production yields, as illustrated in Figure 12, which depicts the rapidity distributions of K$^+$ mesons (upper panel), of K$^-$ mesons (center panel) and of the K$^+$/K$^-$ ratio (lower panel) as measured by the KaoS [47] and FOPI [48,49] collaborations in Ni+Ni collisions at 1.93 A GeV. The measured data are represented by full dots, the open symbols are mirrored at $y_{CM}$ = 0. The lines correspond to BUU calculations with in-medium masses (solid lines) and without in-medium effects (dotted lines) [50]. The calculations without in-medium mass modification overestimate the K$^+$ meson multiplicity and strongly underestimate the K$^-$ yield, as expected for the increased K$^+$ in-medium mass and the decreased K$^-$ in-medium mass.

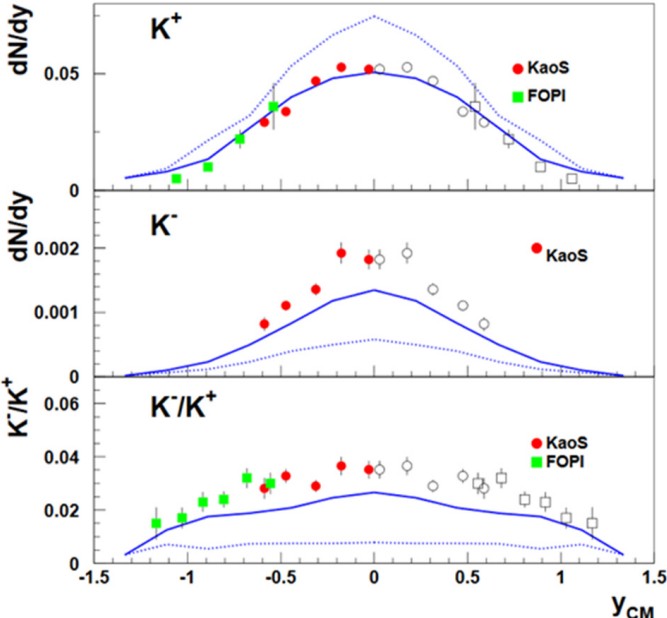

**Figure 12.** Multiplicity density distributions of K$^+$ mesons (upper panel), K$^-$ mesons (center panel) and the K$^+$/K$^-$ ratio (lower panel) for near-central ($b$ < 4.4 fm) Ni+Ni collisions at 1.93A GeV, measured by KaoS (red circles) [47] and FOPI (green squares) [48,49]. The measured data (full symbols) are mirrored at $y_{CM}$ = 0 (open symbols). The data are compared to BUU transport calculations [50]. Solid lines: with in-medium effects. Dotted lines: without in-medium effects. Taken from [33].

## 4. The High-Density Nuclear Matter Equation-of-State

High-energy heavy-ion collisions offer the unique opportunity to study the properties of nuclear matter at high densities in the laboratory. Of fundamental interest is the equation of state (EOS), which is relevant for our understanding of neutron stars, supernova explosions and neutron star mergers. For the temperature T=0, the EOS can be expressed as P = $\rho^2$ d(E/A)/d$\rho$, where P is the pressure, $\rho$ is the density and E/A is the energy per nucleon, which depends on density and isospin, E/A($\rho,\delta$) = E/A($\rho$,0) + E$_{sym}$($\rho$)·$\delta^2$. The first term describes isospin-symmetric matter, while the second term refers to neutron-rich matter, i.e., the symmetry energy times the asymmetry parameter $\delta$ = ($\rho_n−\rho_p$)/$\rho$. For isospin-symmetric nuclear matter, as it is approximately realized in nuclei and heavy-ion collisions, E/A has a minimum at saturation density E/A($\rho_0$,0) = -16 MeV and a curvature parametrized by nuclear incompressibility K$_{nm}$ = 9$\rho^2$·$\delta^2$(E/A)/$\delta\rho^2$. From the experimental study of giant monopole resonances in heavy nuclei, i.e., for saturation density, a value of K$_{nm}$ ($\rho_0$) = 240 $\pm$ 20 MeV has been extracted [51], although somewhat higher values in the range 250 $\leq$ K$_{nm}$ ($\rho_0$) $\leq$ 315 MeV are not excluded [52].

In central heavy-ion collisions at SIS18 beam energies, nuclear densities above 2 $\rho_0$ are reached over a time span of at least 10 fm [53]. According to microscopic transport models, subthreshold K$^+$ production in heavy-ion collisions also exhibits a sensitivity to the

density in the fireball. This is illustrated in Figure 13, which, in the upper panel, illustrates the density reached in a central Au+Au collision at 1A GeV as a function of collision time, while, in the lower panel, the corresponding multiplicities of the created $\Delta$ resonances, pions and $K^+$ mesons are shown. The $\Delta$ resonances are produced in the course of the collision and finally decay into pions, while the $K^+$ mesons are created predominantly at densities above 2 $\rho_0$, as illustrated by the two vertical lines and the blue curve representing the $K^+$ multiplicity in Figure 13. The key mechanism for subthreshold $K^+$ production is the accumulation of the required energy by multiple collisions of particles in the fireball, as described in the previous section. The yield of the created $K^+$ mesons increases with the matter density according to its incompressibility.

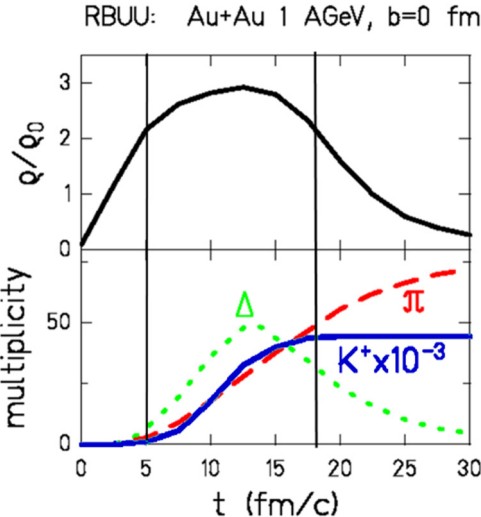

**Figure 13.** Upper panel: Density in the reaction volume as function of time for a central Au+Au collision at 1A GeV as calculated by a RBUU transport model. Lower panel: Multiplicities of produced $\Delta$ resonances (green dotted line), pions (red dashed line) and $K^+$ mesons (blue line) as functions of time.

The black diamonds in the left panel of Figure 14 represent the production cross-sections of $K^+$ mesons measured in Au+Au and C+C collisions as functions of beam energy [54]. The small C+C system serves as a reference, which differs from the large system only in volume and density, hence exhibiting less sensitivity to the EOS. The data are compared to results of QMD transport calculations, assuming a different EOS [55–58]. Calculations without in-medium calculations for C+C collisions overshoot the data by a factor of up to 2 (open symbols), whereas the calculations with in-medium effects are close to the data. It is worthwhile to note that the yield of $K^+$ produced in the light C+C system does not depend on the EOS. The Au+Au data are compared with the results of calculations for a soft EOS ($K_{nm}$ = 200 MeV; blue dots) and a hard EOS ($K_{nm}$ = 380 MeV; cyan squares).

In order to illustrate the influence of the EOS more clearly, the ratio of $K^+$ multiplicities per mass number in Au+Au over C+C collisions is depicted in the right panel of Figure 14. The advantage of this representation is that systematic uncertainties of both data and model calculations are reduced and effects such as in-medium modifications, momentum-dependent interactions, Fermi motion and short-range correlations largely cancel. The data (black diamonds) strongly rise with the decrease in beam energy, i.e., with the increase in "subthresholdness". This trend can be reproduced by different transport calculations when taking into account a soft EOS ($K_{nm}$ = 200 MeV; red symbols), whereas calculations assuming a hard EOS ($K_{nm}$ = 380 MeV; blue symbols) exhibit a much less pronounced energy dependence [57]. The data clearly favor a soft EOS with an incompressibility of $K_{nm}$ = 200 MeV for symmetric nuclear matter at about twice the saturation density.

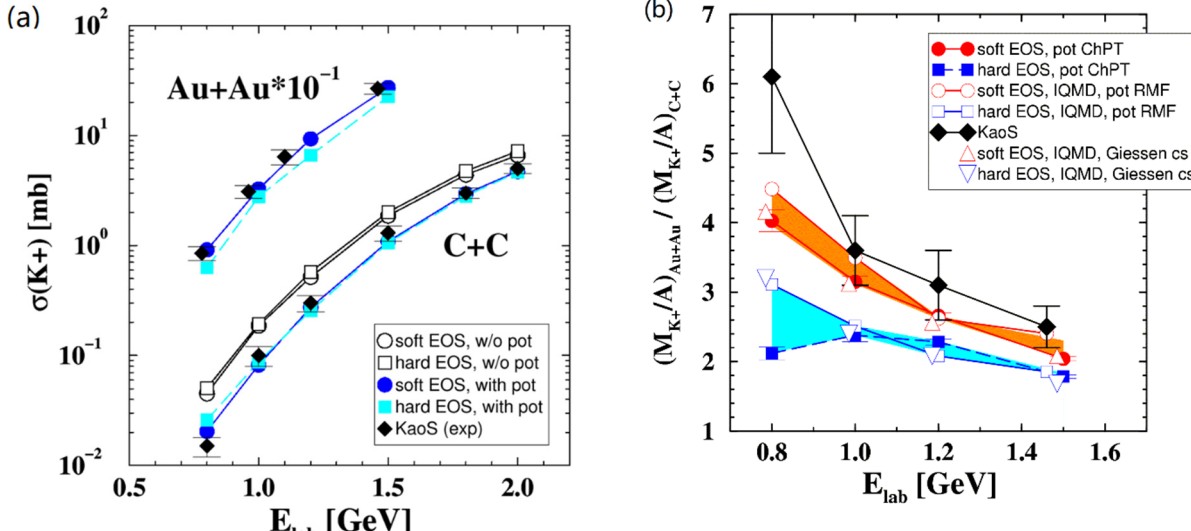

**Figure 14.** (**a**) Production cross-sections of K$^+$ mesons measured in Au+Au and C+C collisions as functions of the projectile energy per nucleon (black diamonds). The data are compared to QMD calculations with (full symbols) and without (open symbols) kaon in-medium modifications, assuming a soft EOS (K$_{nm}$ = 200 MeV; blue dots) or a hard EOS (K$_{nm}$ = 380 MeV; cyan squares). (**b**) Ratio of the K$^+$ multiplicity per mass number in Au+Au over C+C collisions as a function of beam energy. The data are compared to different QMD calculations assuming a soft EOS (K$_{nm}$ = 200 MeV; red symbols) or a hard EOS (K$_{nm}$ = 380 MeV; blue symbols). Taken from [57].

The finding of the KaoS collaboration concerning the EOS was corroborated by results of the FOPI collaboration at GSI, which measured the elliptic flow of protons and light fragments in Au+Au collisions at beam kinetic energies from 0.4A to 1.5A GeV [59]. This observable is very sensitive to the EOS, as the collective flow of nucleons is driven by the pressure gradient in the reaction volume [12]. The FOPI flow data could be well reproduced by IQMD transport calculations when assuming a nuclear incompressibility of K$_{nm}$ = 190 $\pm$ 30 MeV and taking into account momentum-dependent interactions [59].

However, in order to contribute to our understanding of neutron stars, the symmetry energy E$_{sym}$ also has to be determined. According to transport models, E$_{sym}$ can be extracted from the elliptic flow of neutrons and protons generated in heavy-ion collisions [60]. Such a measurement has been pioneered at GSI [4] and recently repeated with an upgraded setup by the ASY–EOS collaboration at GSI by measuring the elliptic flow of neutrons and of charged particles in Au+Au collisions at a beam energy of 0.4A GeV [61]. Using a UrQMD transport code, values for the symmetry energy could be extracted from the comparison of the measured neutron flow and the flow of charged particles up to densities of 2 $\rho_0$, where E$_{sym}$ reaches a value of 55 $\pm$ 5 MeV. When adding the measured E$_{sym}$ distribution to the results for the EOS of symmetric matter as measured by the KaoS and FOPI experiments, one obtains the first experimental EOS for neutron matter extracted from heavy-ion data. The experimental EOSs for both neutron matter (upper green area) and symmetric nuclear matter (lower green area) are shown together with different theoretical calculations in Figure 15, which depicts the binding energy as a function of the density [62].

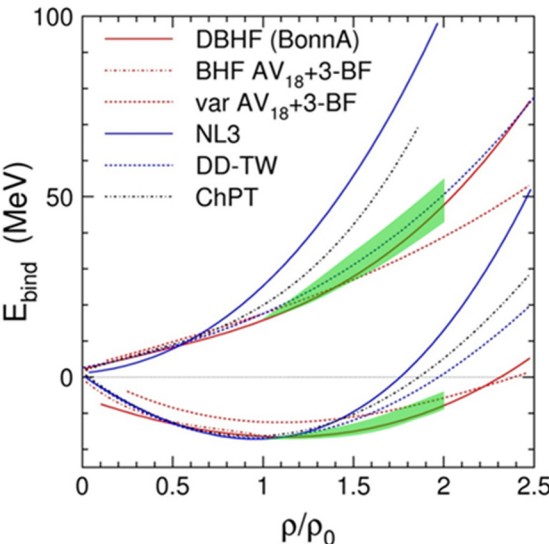

**Figure 15.** Binding energy as a function of nuclear matter density in units of $\rho_0$. The lines represent the results of various calculations for neutron matter (upper curves) and for symmetric matter (lower curves) [62]. Lower green area: EOS for symmetric matter as extracted from data of the KaoS [54,55] and FOPI [59] experiments. Upper green area: Symmetry energy $E_{sym}$ as extracted from the data of the ASY–EOS experiment [61] added to the experimental EOS for symmetric matter (see text).

In order to contribute to our understanding of the structure of neutron stars and of the dynamics of neutron star mergers, the EOS has to be determined up to 5–6 times the saturation density [63]. Such densities are reached in heavy-ion collisions with beam kinetic energies between 5A and 10A GeV [64]. The first investigations of the EOS in heavy-ion collisions at these beam energies were performed at the AGS at BNL, where the collective flow of protons was studied in Au+Au collisions at energies between 2 and 8A GeV [11]. However, the interpretation of the data by relativistic transport model calculations was not very conclusive; while the transverse flow data could be reproduced under the assumption of a soft EOS ($K_{nm}$ = 210 MeV), the elliptic flow data seem to support a stiff EOS ($K_{nm}$ = 300 MeV) [12]. Figure 16 depicts the result of the analysis of the AGS proton flow data as a grey shaded area, plotted as pressure versus density. The blue and red line correspond to a soft and a hard EOS, respectively [12]. The yellow area corresponds to the EOS extracted from the FOPI and KaoS measurements as discussed above. In conclusion, Figure 16 represents the present constraints for the EOS of symmetric nuclear matter above the saturation density, illustrating that the analysis of the heavy-ion data by microscopic transport models find a soft EOS up to densities of about 2 $\rho_0$, whereas, at higher nuclear densities, only very soft or very hard EOSs are excluded.

The uncertainties concerning the high-density EOS of symmetric nuclear matter as illustrated in Figure 16 clearly call for improved data and calculations. Measurements of a particle flow at beam energies around 3A GeV have been already performed by the STAR experiment at BNL in the fixed target mode [26]. In addition to the flow measurements, sub-threshold particle production may also be a very sensitive probe of the high-density EOS of symmetric nuclear matter. At beam energies between 2A GeV and 10A GeV, multi-strange (anti-hyperons) are expected to be the most promising candidates in this respect. This has been studied using the new Parton-Hadron-Quantum-Molecular Dynamics (PHQMD) transport code by simulating hyperon production in central Au+Au collisions at a beam energy of 4A·GeV. According to preliminary calculations, the yield of $\Xi^{\pm}$ and $\Omega^{\pm}$ hyperons clearly depends on the stiffness of the EOS for symmetric nuclear matter [65].

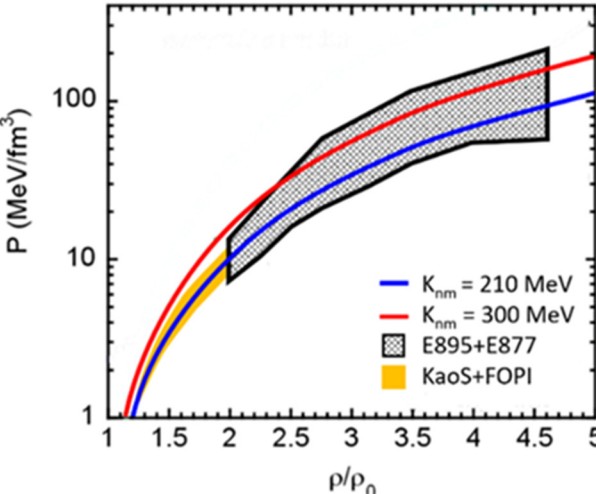

**Figure 16.** EOS of symmetric nuclear matter expressed as pressure versus baryon density. Grey hatched area: constraint from proton flow data taken at AGS [11,12]. Yellow area: Constraint from fragment flow and kaon data taken at GSI [54,55,59]. Red line: Hard EOS. Blue line: Soft EOS [12].

In order to contribute to our understanding of neutron stars and neutron star mergers, the symmetry energy $E_{sym}$ also has to be experimentally studied at densities up to about 5 $\rho_0$. Possible observables sensitive to $E_{sym}$ include the flow of neutrons compared to the flow of charged fragments, as measured by the ASY-EOS collaboration at GSI-SIS18, and the ratio of particles with opposite isospin, which reflect the density distributions of protons and neutrons. For example, the $\pi^-/\pi^+$ ratio has been investigated as a probe for the $E_{sym}$, but it has turned out that the sensitivity to $E_{sym}$ vanishes at beam energies well above the pion production threshold. In addition, the pion ratio depends on the $\Delta(1232)$ in-medium potential, which is not well known [66]. A more promising observable might be particles with higher production thresholds and different isospin projections $I_3 = \pm 1$, such as $\Sigma$ hyperons. The $\Sigma^-$ (dds)/$\Sigma^+$(uus) ratio could be used as a proxy of the n(ddu)/p(uud) ratio [67].

The EOS for symmetric nuclear matter will be investigated in the near future by the upgraded BM@N experiment at the JINR-Nuclotron, where both proton flow and hyperons will be measured in Au+Au collisions at beam at energies of up to 3.8A GeV [68]. The CBM experiment at FAIR-SIS100 will study these observables for higher densities, i.e., in Au+Au collisions at beam energies up to 11A GeV [69]. The $E_{sym}$ at neutron star core densities will be explored by measuring the $\Sigma^-/\Sigma^+$ ratio for different collision systems and beam energies.

Information on the EOS of neutron matter can also be obtained by the analysis of astronomical observations, such as simultaneous measurements of radii and masses of neutron stars by the NICER experiment [70] and the detection of gravitational waves emitted from mergers of compact stars [71]. A recent theoretical study demonstrates that the combination of astronomical observations and results of laboratory experiments reduces the uncertainties of the complementary approaches and provides an improved experimental constrain of the high-density EOS [72].

## 5. Summary

The KaoS collaboration performed pioneering measurements of subthreshold strangeness production in heavy-ion collisions at the SIS18 accelerator at GSI. The experiments were performed with a double-focusing dipole magnetic spectrometer, equipped with detectors for particle identification and for the determination of collision centrality and orientation of the reaction plane. The measured data provide information on in-medium properties of kaons and on the equation-of-state of dense symmetric nuclear matter. Together with the data on collective flow of protons and light fragments measured by the FOPI

collaboration at GSI, the up-to-date KaoS data constrain the EOS of symmetric matter up to densities of twice the saturation density. Future experiments planned at NICA and FAIR will extend our knowledge relative to the high-density EOS. The combined information from the interpretation of astronomical observations and laboratory experiments will open a truly multi-messenger era of fundamental studies of matter at neutron star core densities.

**Funding:** The author acknowledges support from the Europeans Union's Horizon 2020 Research and Innovation Programme under grant agreement No. 871072 and from the Russian Foundation for Basic Research (RFBR) according to the research project No. 18-02-40086 by the Ministry of Science and Higher Education of the Russian Federation, Project "Fundamental properties of elementary particles and cosmology" No. 0723-2020-0041.

**Data Availability Statement:** Not applicable.

**Conflicts of Interest:** The authors declare no conflict of interest.

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
