# Peer review of "Pioneering the Equation of State of Dense Nuclear Matter with Strange Particles Emitted in Heavy-Ion Collisions: The KaoS Experiment at GSI"

_2571-712X, doi:10.3390/particles5010003_

Round 1

Reviewer 1 Report

The manuscript  of  P. Senger `` Pioneering the equation-of-state of dense nuclear matter with strange particles emitted in heavy-ion collisions: The KaoS experiment at GSI’’  reviews   measurements done first  at the LBL in Berkeley in the early 1980’s   and later by KaoS experiment at GSI.  After focusing on pion production  the experimental results on subthreshold kaon production together with their theoretical interpretations are considered and  a  brief outlook towards future experiments at higher densities is done.  

The author  is one of the most  known  experimentalists  in  heavy-ion collision physics.  Many important results were performed by his group at GSI during many years. The manuscript of  P. Senger   is clearly written and its publication will be helpful for the audience  interested in heavy-ion collision physics. Thereby, I recommend publication of the manuscript. Only on consideration of the author I would suggest   a   minor revision, see below.

Lines 34-36:  The Streamer Chamber collaboration at LBL measured the excitation function of pion production in La+La collisions, and estimated  from the pion yields the fireball temperature and the potential part of the compressional  energy, indicating a stiff nuclear matter equation-of-state (EOS) [2].

(Referee). I would suggest to soften the statement. Stiff  EoS is quite not necessary to appropriately explain these  data, especially such strongly stiff as suggested in [2].  For example see Phys.Repts. 192 (1990) 179, Figs.  11.13-11.15,  the GSI RMF model (Phys. Lett. B201, 11 (1988)) using  soft EoS  with incompressibility K=210 MeV and with additionally included temperature-density pion in-medium effects was successfully employed.  Therefore both the Bevalac data and GSI ones are reasonably described using the same incompressibilities plus in-medium T-n dependent pion effects.

Lines 243-244: In conclusion, the experimental results confirm the picture, that hard pions leave the fireball early, whereas soft pions freeze-out late.

(Referee). I would suggest to soften the statement.  It is in contradiction  to the estimates and experimental results on pion path lengths, cf. Nucl. Phys. A555 (1993) 293 Fig 2. In mentioned above and subsequent works the pion spectra were successfully described using that  hard pions leave fireball at freeze-out and  soft pions (with k<m_\pi) leave fireball earlier.

Lines 351-354: According to model calculations [41], the slightly  repulsive K+N in-medium potential results in a K+ effective mass moderately increasing  with nuclear density, whereas the strongly attractive K−N potential leads to a K− effective  mass which considerably decreases with increasing density. The latter effect may also result in a reduced K− absorption cross section in nuclear matter.

(Referee). Kaon baryon interaction is described similarly  to the \pi N one. The  kaon-baryon \Sigma term is larger than pion-nucleon one and therefore the s-wave attractive antikaon potential is stronger, but the p-wave interactions are very similar, see e.g. in Intern. Journ. of Modern Phys. E, 5 (1996) 316. I would  suggest to mention importance of the p-wave effects. They are not less strong compared to  the  s-wave potential ones.

Lines 383-385:  From the analysis of measurements of  the giant monopole resonance in heavy nuclei, a value for the incompressibility of Knm(ρ0) = 230 ± 10 MeV has been extracted, which corresponds to a soft EOS at saturation density  [48].

(Referee). Actually experimental error-bars are essentially broader, lowest error-bars give  K=240\pm 20 MeV as presented in EPJA 30 (2006) 23, whereas Phys. Rev.C89 (2014) 044316 gives K=250-315 MeV.  Most difficult is to marry HIC flow result shown in your Fig. 16 that requires rather soft EoS  for iso-symmetrical matter  and experimental result, Nature Astronomy 4 (2020) 72, on the maximum mass of neutron stars, M> 2 M_sol, that requires stiff EoS for neutron star matter. Note  that hyperon and  delta-isobar appearance  in neutron stars  still softens EoS. Up to now only few models of EoS could  describe  these results simultaneously, cf.  Nucl. Phys. A  961 (2017) 106.

Author Response

Dear Reviewer,

thank you very much for the comments. Please find attached my answers.

Reviewer 2 Report

The paper surveys nicely and in a comprehensive manner the big step made by the Kaon Spectrometer Collab. (together with FOPI) from Bevalac to HADES-GSI towards HADES-FAIR and CBM-FAIR and NICA and other forthcoming installations, aimed at investigating compressed nuclear matter and its constituents. Both instrumentation aspects and physics interpreations are summarized in a balanced representation. The connection to modern astrophysics issues is clearly highlighted.

I recommend publication of this paper after some minor corrections.

  • l.83: replace "Exactly" --> "Consistently" since the SHM fit results have some spread depending on the selected hadron species sample (have in mind the "proton puzzle", for instance)
  • l.166: provide angular ranges for the colored bands
  • often "Delta" is not displayed, e.g. l.208, or angles look strange, e.g. l.225 and 233
  • l.256: is really "row" meant, not "column"?
  • l.276: are "first-chance collisions" meant or really all "binary nucleon-nucleon collisions"?
  • l.281: define better "Q-value"
  • l.282: replace "inset" --> "legend"
  • Fig. 8: explain meaning of the curves
  • bibliography: often order/bf-print of year and volume is messed up
  • missing title in [54]
  • missing authors and title in [67]

Author Response

Dear Reviewer,

thank you very much for your comments which I have taken into account, see attached file.

Reviewer 3 Report

This is a review of old research. Therefore, I shied away from answering the question "Originality / Novelty". 

Author Response

Dear Reviewer,

thank you for your answers.

Reviewer 4 Report

The author points out the significance of understanding the nuclear matter equation-of-state (EOS) and the degrees-of-freedom of matter at high densities and temperatures. The author reviews the experimental measurements on subthreshold kaon production jointly with the theoretical predictions and provides a view for the upcoming experiments at higher densities.

I will recommend the review article for publication with suggestions that I will leave them up to the author.

1) In the introduction you can add more references for example "Lineine-104: fixed targets at STAR"
2) som some of the plots if the data is published I will advise the author  to replot them as "Fig.6,10,13 and 16"

Author Response

Dear Reviewer,

thank you very much for the comments. I have included a reference to STAR fixed target measurments.